# Gut Microbiome in Children from Indigenous and Urban Communities in México: Different Subsistence Models, Different Microbiomes

**DOI:** 10.3390/microorganisms8101592

**Published:** 2020-10-16

**Authors:** Andrés Sánchez-Quinto, Daniel Cerqueda-García, Luisa I. Falcón, Osiris Gaona, Santiago Martínez-Correa, Javier Nieto, Isaac G-Santoyo

**Affiliations:** 1Laboratorio de Ecología Bacteriana, Instituto de Ecología, UNAM, Mexico City 04510, Mexico; falcon@ecologia.unam.mx (L.I.F.); ogaonap@gmail.com (O.G.); 2Consorcio de Investigación del Golfo de México (CIGOM), Departamento de Recursos del Mar, Centro de Investigación y de Estudios Avanzados del Instituto Politécnico Nacional, Unidad Mérida, Yucatan 97310, Mexico; dacegabiol@ciencias.unam.mx; 3Instituto de Ecología, Campus Yucatán, Parque Científico y Tecnológico de Yucatán, Mérida 97302, Mexico; 4Neuroecology Lab, Facultad de Psicología, UNAM, Mexico City 04510, Mexico; shantimtz@gmail.com; 5Laboratorio de Aprendizaje y Adaptación, Facultad de Psicología, UNAM, Mexico City 04510, Mexico; janigu@unam.mx

**Keywords:** intestinal microbiome, children’s microbiota, diet, Westernized, non-Westernized, lifestyle, microbial diversity, human health

## Abstract

The human gut microbiome is an important component that defines host health. Childhood is a particularly important period for the establishment and development of gut microbiota (GM). We sequenced the 16S rRNA gene from fecal samples of children between 5 and 10 years old, in two Mexican communities with contrasting lifestyles, i.e., “Westernized” (México City, *n* = 13) and “non-Westernized” (Me’phaa indigenous group, *n* = 29), in order to characterize and compare their GM. The main differences between these two communities were in bacteria associated with different types of diets (high animal protein and refined sugars vs. high fiber food, respectively). In addition, the GM of Me’phaa children showed higher total diversity and the presence of exclusive phyla, such as Deinococcus-Thermus, Chloroflexi, Elusimicrobia, Acidobacteria, and Fibrobacteres. In contrast, the children from México City showed less diversity and the presence of Saccharibacteria phylum, which was associated with the degradation of sugar compounds and was not present in the samples from Me’phaa children. This comparison provided further knowledge of the selective pressures affecting microbial ecosystemic composition over the course of human evolution and the potential consequences of pathophysiological states correlated with Westernization lifestyles.

## 1. Introduction

The human gastrointestinal tract is colonized by an abundant and diverse assemblage of microorganisms collectively known as the gut microbiota (GM) which impact host physiology [1]. GM are composed of more than 2000 genera with an incredible diversity of functions that influence host health [2,3]. GM synthesize a huge number of peptides (more than are encoded in the human genome) [4] and participate in the biosynthesis of vitamins, fermentation of dietary polysaccharides, absorption of ions, and regulation of a number of host metabolic pathways [4,5]. Moreover, GM secrete antimicrobial peptides that help to maintain homeostasis [6] and regulate the development, function, and adaptation of the innate immune system [3]. To date, at least 50 human pathologies have been associated with changes in the abundance, composition, and dynamics of GM [7]. These pathologies are associated with intestinal issues such as bowel diseases or gastrointestinal cancer, and also with autoimmune disease, metabolic syndromes, and neurological pathologies [7,8]. The “core” human GM have been reported to consist of about 14 genera, including *Bacteroides*, *Bifidobacterium*, *Enterococcus*, and *Prevotella* [9]. Nevertheless, more than 2000 genera have been characterized to date in human GM, and their abundance and composition can vary greatly among individuals, resulting in up to 80% genetic dissimilarity of the GM from one person to another [2,10].

Interestingly, although host-associated microbes are presumably acquired from the environment, GM are surprisingly different from the environment surrounding the host [10]. This suggests a coevolutionary process between humans and our GM taxa, which adapt to the internal environment of a particular host [11]. Individual lifestyle has been identified as one of the most important factors determining the composition, abundance, and stability of the intestinal microbiome [12]. In particular, sociocultural practices such as cesarean section versus vaginal delivery during birth, early life feeding, access to allopathic medications, and diet have been studied the most [2]. The other important factor is the host’s age. It is clear that children’s GM differ from adult GM in several important respects. For instance, compared to adults, children’s microbiota are enriched with *Bifidobacterium* spp. and *Faecalibacterium* spp., which play an important metabolic role in development during childhood [13]. Over at least the first three to five years of life, the composition of GM is in flux; the diversity of GM increases and converges toward an adult-like microbiota, with characteristics such as the overall number of taxa and functional genes and becomes increasingly similar to the adult GM over time [14,15,16]. This period before the stabilization of the microbiota is critical for children’s growth and development, since alteration of the GM can influence adult health [14,17]. While the GM has been extensively studied in both adults and in children between 0 and 2 years of age [18,19], there is very little information about the GM of older children, despite evidence that their GM are more dynamic than previously considered and continue to change in important ways even at ages very close to adulthood [19]. In addition, the vast majority of GM studies have been carried out in populations with a so-called “Westernized lifestyle” [18]. To date, fewer than 15 studies have characterized the GM in children from populations that do not conform to the “Westernized lifestyle” [12], including hunter-gatherers such as the Hazda of Tanzania [20]; subsistence farmers in Bassa, Nigeria [21]; and Amerindians in South America [22].

Evaluating populations with diverse lifestyles is fundamental because it allows us to explore how lifestyle practices impact the structure of human GM, and to acquire insights into how GM have adapted with changes over the course of human evolution from a hunter-gatherer lifestyle, through small scale agriculture, to a post-industrial Westernized lifestyle [23], and how those adaptations can affect human health [23].

The information obtained from the few studies in non-Westernized lifestyles indicates that a shift to a Westernized lifestyle can decrease the abundance of different taxonomic groups due to a decrease in consumption of microbiota accessible carbohydrates (MACs) in the diet [12]. These groups are known as VANISH (volatile or associated negatively with industrialized societies of humans) and include species from the families Prevotellaceae, Spirochaetaceae, and Succinivibrionaceae. These VANISH organisms are capable of degrading complex plant-derived carbohydrates rich in fiber [12] by encoding different types of carbohydrate-active enzymes (CAZyme) such as glycoside hydrolase [24]. At the same time, a Westernized lifestyle can decrease phylogenetic richness, and GM of people from Westernized lifestyles are more likely to contain bacteria from a group of taxa known as BIoSSUM (bloom or selected in societies of urbanization/modernization) that includes members of the Bacteroidaceae, Enterobacteriaceae, and Verrucomicrobiaceae families [12,24].

In México, there are at least 56 independent indigenous groups whose lifestyle practices differ from the typical “Westernized lifestyle” to varying degrees [25]. Among these, the Me’phaa people from a region known as “Montaña Alta” in the state of Guerrero, is one of the groups whose lifestyle differs most strongly from the “Westernized lifestyle” typical of more urbanized areas [26,27,28]. The Me’phaa is a prehispanic indigenous group, where people live in communities composed of fifty to eighty families, each with five to ten family members. Most people only speak their native language [26], and they mainly live by subsistence farming of legumes including beans and lentils, and the only grain cultivated is corn. Wild edible plants are also collected, and some fruits and vegetables are cultivated in garden plots [29]. Animal protein is acquired by hunting and raising some fowl, but meat is consumed almost entirely during special occasions and is not part of the daily diet [29]. All food resources are produced locally, cultivated and harvested near the community [29]. Ninety-eight percent of births are by vaginal delivery, and children are breastfed until the age of two. There is almost no access to allopathic medications, and there is no health service, plumbing, or water treatment [27]. Water for washing and drinking is obtained from small wells. These communities have among the lowest income in the country and the highest indices of child and adult morbidity and mortality [30]. In consequence, the inhabitants of this region have a contrasting lifestyle as compared with other regions in the country, such as México City, the most urbanized city of the country and the fifth most populous city in the world [31].

The aim of this study was to explore whether these contrasting lifestyles influence the ecosystem dynamics of GM in childhood, an important age for the establishment and development of GM. To evaluate this, we determined the GM abundance and composition and interpopulation differences of two groups of 5–10-year-old Mexican children with strongly contrasting lifestyle practices, i.e., children from México City with a “Westernized” lifestyle, and children from the Me’phaa ethnic group with a “non-Westernized” lifestyle.

## 2. Materials and Methods

### 2.1. Study Site

The feces of children from México City (18.102″ W 19°12′36.36″ W) and two Me’phaa communities, Plan de Gatica (17°7′ 49.5552″ N 99.7′, EASL 510 m) and El Naranjo (17°9′ 54.0036″ N 98°57′, 50.9832″ W, EASL 860 m) were obtained. The distance between the two indigenous communities is almost 30 km, and the socioeconomic and cultural patterns are similar [29].

In this work, the “Westernized” population was represented by children that inhabited the south of México City and that were part of a federal pedagogical program at the National Pedagogical University (Figure 1). This population corresponded to a medium-high economic level with abundant diets characterized by high animal protein consumption, refined vegetable oils, cereal grains, and sugars (e.g., soda, biscuits, snacks, etc.), as well as low fiber and vegetables intake.

### 2.2. Sample Collection

Fecal samples were obtained from children between 5 and 10 years old, in the Me’ phaa community and México City; 33 children (15 males and 18 females) and 13 children (four males and nine females) were collected, respectively. Despite the limitation of not being a vast number of samples, other authors have previously used this approximation [21]. Each participant collected a fecal sample in a sterilized plastic jar. Each jar had a unique nomenclature designated to the participant written on the lid. All fecal samples were frozen with liquid nitrogen in a posterior storage at −20 °C, until DNA extraction. Before the DNA extraction, fecal samples of approximately 100 µL were collected with a pipette tip and placed in a 1.5 milliliter sterile microtube.

In addition to fecal samples, the study included anthropometric measurements (i.e., height, weight, and BMI) from all participants (Appendix A
Appendix A), as well as a questionnaire applied to the corresponding mothers about the children’s nutritional status and information on pregnancy, childbirth, and length of breastfeeding. For the Me’phaa community, the assistance of a translator was needed, since this community does not speak the Spanish language.

### 2.3. Fecal DNA Extraction

Each sample (~100 µL of fecal DNA) was extracted using the DNeasy Blood & Tissue kit (Qiagen, Valencia, CA, USA), according to the manufacturer’s instructions. DNA was resuspended within 30 µL of molecular grade water and stored at −20 °C until PCR amplification.

### 2.4. 16S rRNA Gene Amplification and Sequencing

The hypervariable V4 region of the 16S rRNA gene was amplified with universal bacterial/archaeal primers 515F/806R [12,32], following the procedures reported by Caporaso et al. (2012) [33]. The PCR mix was done in 25 µL reactions in triplicate per sample as follows: 2.5 µL Takara ExTaq PCR buffer 10×, 2 µL Takara dNTP mix (2.5 mM), 0.7 µL bovine serum albumin (BSA, 20 mg mL^−1^), 1 µL primers (10 µM), 0.125 µL Takara Ex Taq DNA Polymerase (5 U µL^−1^) (TaKaRa, Shiga, Japan), 2 µL of DNA, and 15.67 µL nuclease-free water. The PCR protocol included an initial denaturation step at 95 °C (3 min), followed by 35 cycles at 95 °C (30 s), 52 °C (40 s), and 72 °C (90 s), followed by a final extension (72 °C, 12 min). Triplicates were pooled and purified using the SPRI magnetic bead, AgencourtAMPure XP 214 PCR purification system (Beckman Coulter, Brea, CA, USA). Characterization of the fecal purified 16S rRNA fragments (∼20 ng per sample) was sequenced on an IlluminaMiSeq platform (Yale Center for Genome Analysis, New Haven, CT, USA) and ∼250 bp paired-end reads were generated. All sequences obtained were uploaded to the NCBI database under the Bioproject number PRJNA593240.

### 2.5. Analysis of the Sequence Data

The paired-end 2 × 250 reads were processed in QIIME2. The reads were denoised with the DADA2 plugin to resolve the amplicon sequence variants (ASVs) [34]. Forward and reverse reads were both truncated at 200 pb, and chimeric sequences were removed using the “consensus” method. Representative ASV sequences were taxonomically assigned using the “classify consensus-vsearch plugin” [35], using the SILVA 132 database as a reference [36]. An alignment was performed with the MAFFT algorithm [37]. After masking positional conservations and gap filtering, a phylogeny was built with the FastTree algorithm [38]. The abundance table and phylogeny were exported to the R environment to perform the statistical analysis with the phyloseq [39] and ggplot2 packages. Plastidic ASVs were filtered out of the samples, which were rarefied to a minimum sequencing effort of 21,000 reads per sample (Appendix A
Appendix A). A Venn diagram was plotted with the ASVs by location using the vennDiagram function of the limma library in R [40]. The total diversity (alpha diversity) of the ASVs was calculated using Faith’s phylogenetic diversity index (PD), Shannon diversity index, and observed ASV’s.

### 2.6. Statistics Analyses

To determine whether or not alpha diversity is different between community and sex, we performed a Welch two sample *t*-test to evaluate Faith’s PD index and Shannon index; a Wilcoxon rank-sum test with continuity correction to evaluate observed ASVs. Comparisons were done in all possible combinations between sex (i.e., male and female) and community (indigenous and Westernized). Beta diversity analysis between sex and community was estimated by computing weighted, unweighted UniFrac and Bray–Curtis distances. Statistical differences were determined by a permutational multivariate analysis of variance using distance matrices (PERMANOVA). Additionally, we performed a differential abundance analysis with the DESeq2 library [41] to determine the main discriminant ASVs between community and sex.

We additionally explored the particular differences of the most characterized VANISH and BIoSSUM families, i.e., Prevotellaceae, Spirochaetaceae, and Succinivibrionaceae as the VANISH group, and Bacteroidaceae, Enterobacteriaceae, and Verrucomicrobiaceae as the BIoSSUM group [12]. Hence, we performed a zero inflated beta regression model (ZIBM) considering community and sex of the children as additive predictors. If abundance was 0 in ≥90% of the studied samples, a Wilcoxon rank-sum test was performed instead of ZIBM. Statistically significant comparisons in all alpha and beta diversity analyses were considered, with *p* < 0.05. *p*-values in all multiple comparisons were adjusted by the Holm’s method [42]. Comparative statistical analysis was performed in R 3.5.0 [43] implementing the following packages: vegan [44], MASS [45] and GAMLSS [46], stats [43].

### 2.7. Ethics

All procedures for testing and recruitment were approved (25 September 2017) by the National Autonomous University of México Committee on Research Ethics (FPSI/CE/01/2016) and run in accordance with the ethical principles and guidelines of the Official Mexican Law (NOM-012-SSA3-2012). All participants read and signed a written informed consent. Additionally, we received signed approval from all participants (or their parents) who were photographed for the manuscript.

## 3. Results

### 3.1. Microbial Taxonomic Comparison from México City and Me’phaa Communities

This study generated a dataset of 42 fecal samples, i.e., 29 children (ages 5–10 years) from the Me’phaa indigenous group (Montaña Alta de Guerrero) and 13 children (ages 5–10 years) from México City. A total of 336,000 sequences were recovered after performing the quality filtering and removing chimeras. Firmicutes and Bacteroidetes were the dominating phyla in both populations (Figure 2A,C–H). Firmicutes, Bacteroidetes, and Actinobacteria were higher in children from México City than those from the Me’phaa (77.10–68.04%, χ² = 7385, *p* < 0.001; 14.84–12.07%, χ² = 1271, *p* < 0.001; and 5.34–2.44%, χ² = 4826, *p* < 0.001, respectively) (Figure 2H). In contrast, some phyla were only present within the indigenous community including Deinococcus-Thermus (0.079%), Chloroflexi (0.01%), Elusimicrobia (0.01%), Acidobacteria (0.0071%), Fibrobacteres (0.004%), and only Saccharibacteria (0.0003%) were present in urban GM (Appendix A) (Figure 2H).

Among the males from the city, Firmicutes (66.5%) were found in a lower proportion in relation to Bacteroidetes (26.24%) abundance, in contrast to females from the city and children from the indigenous population (Figure 2C–F). Moreover, females from México City comprised the group that presented the highest proportion of Firmicutes (81.7%) (Figure 2E). In addition, the abundance of Tenericutes (11.50% and 7.55%) and Proteobacteria (5.34% and 3.05%) was higher within females and males of the indigenous population as compared with urban children’s GM (Tenericutes, female city 0.70%, χ² = 21504, *p* < 0.001 and male city 0.27%, χ² = 17493, *p* < 0.001; Proteobacteria, female city 0.80%, χ² = 8197, *p* < 0.001 and male city 0.61%, χ² = 5162, *p* < 0.001, respectively) (Figure 2A,C–H). Overall, the diversity in the indigenous children was greater than for urban children (Figure 3; Appendix A). The Venn diagram (Figure 2B) revealed that only a quarter (23.64 %) of the total ASVs was shared between the indigenous and urban populations sampled. Specific ASVs from the city and the indigenous community (25% and 51.3%, respectively) indicated that the shared diversity is lower than specific ASVs from each location (Figure 2B). For bacterial composition at the family level, please refer to Appendix A.

### 3.2. Alpha Diversity, Clustering Ordination, and Family/Genus Comparison of GM in Children from México City and Me’phaa Communities

Faith´s PD index was statistically significant between communities (*t* = −3.54, *p*-value < 0.01), but not between sexes (*t* = −1.75, df = 29.70, *p*-value = 0.092). Children from the Me’phaa community showed greater diversity as compared with children from México City (Figure 3A). A similar relation was found implementing the Shannon index, the Me’phaa community had greater alpha diversity than children from México City (Figure 3B, *t* = −1.87, df = 29.8, *p*-value < 0.06), which was more accentuated between sexes; males showed greater diversity than females in both communities (*t* = −2.1983, df = 28.075, *p*-value = 0.03, Figure 3B). The difference between sexes was also present in the number of observed ASVs (W = 134.5, *p*-value = 0.04), although there were no significant differences between communities (Figure 3C, W = 131.5, *p*-value = 0.12).

The GM composition in children shows a clear separation between the urban and indigenous communities when unweighted Unifrac was obtained (Figure 3D), PERMANOVA, F = 5.72, R2 = 0.13, *p* < 0.001). However, this separation was not observed when weighted Unifrac was implemented (Appendix A, PERMANOVA, F = 2.79, *p* = 0.08). Both distance metrics were independent of sex (Adonis for unweighted, F = 1.16, *p* = 0.21; for weighted, F = 1.89, *p* = 0.24). Bray–Curtis dissimilarity was also evident between communities (Figure 3E, PERMANOVA, F = 6.37, R2 = 0.14, *p* < 0.01), regardless of sex (PERMANOVA, F = 1.41, *p* = 0.09).

Each location had specific ASVs, detected in the log2 fold-change analysis (Figure 4). ASVs of *Akkermansia*, *Ruminococcus 1*, *Coprostanoligenes*, and *Phascolarctobacterium* genera were mostly associated with the México City children. In contrast, *Prevotella 7* and *9*, *Treponema 2*, *Catenibacterium*, *Christensenellaceae R-7 group*, *Faecalibacterium*, *Ruminococcaceae UCG-009*, and *Ruminococcaceae UCG-014* were associated with the Me’phaa community.

### 3.3. The Contrasting Relationship between VANISH and BloSSUM Groups between Indigenous and City Children

According to Sonnenburg and Sonnenbgurg (2019), there are differences in the abundance of VANISH taxa (volatile or associated negatively with industrialized societies of humans) and those positively associated with societies of urbanization/modernization (BIoSSUM). For this manuscript, we considered community and sex of the children to be predictors of the following families: Bacteroidaceae, Enterobacteriaceae, Verrucomicrobiaceae, Prevotellaceae, Spirochaetaceae, and Succinivibrionaceae (Figure 5) [12].

Children’s GM from México City exhibit a greater abundance in two of three BIoSSUM groups, i.e., Bacteroidaceae (*t* = −4.78, *p* < 0.01) and Verrucomicrobiaceae (*t* = −2.81, *p* < 0.01) as compared with the Me’phaa community (Figure 5D,E). This was independent of the sex of the children (*t* = −0.32 and *p* = 0.71). Although the Enterobacteriaceae family was more abundant in the Me’phaa community (0.03 vs. 0.01), the difference was not statistically significant (*t* = 1.86, *p* = 0.07, Figure 5F). In contrast, the VANISH groups Prevotellaceae (*t* = 2.97, *p* < 0.01), Spirocheataecae (W = 81, *p* < 0.01), and Succinivibrio (W = 104, *p* < 0.01) were more abundant in children from the Me’phaa community than in children from México City (Figure 5A–C).

## 4. Discussion

Our results suggest that GM are affected by contrasting lifestyles such as those observed in the Me´phaa indigenous community that mirror the traditional pre-Columbian conditions, versus a typical Westernized society. Previous reports have illustrated similar differences, suggesting a general impact of lifestyle practices [23,47,48,49,50,51]. Similar to other studies in “non-Westernized” populations from Africa and South America [18,22], both GM richness and abundance were lower in México City than in the Me’phaa community (Figure 3), which was reflected as a loss of microbial organisms and genes [23,47]. The higher richness and diversity of fecal microbiota could be due in part to the greater intake of plant-derived carbohydrates from a diet rich in fiber and grains that is common in Me’phaa communities [12,52]. Similar to the Mossi ethnic group from Burkina Faso [18], the Me’phaa diet is low in fat and animal protein [53]. Furthermore, the use of antibiotics as well as high fat diets is often associated with a reduction in biodiversity within Westernized populations [54], as is a shift from Bacteroidetes to Firmicutes [55,56]. In addition to these factors, GM are also driven by other factors that contrast between the two communities explored, such as sanitation, social behavior, climate, type of birth, breast feeding, parental care, etc. [12,18,22,23,48,49,50,51,57].

Firmicutes, Bacteroidetes, Actinobacteria, and Verrucomicrobia are common phyla reported for human gut diversity in both Westernized and non-Westernized populations [58]. With the exception of Verrucomicrobia, these taxa were also highly represented in the two communities. This suggests their presence as part of the general “core” human GM (Figure 2B). Despite these broad similarities, there were strong contrasts in the microbial diversity and abundance of other less represented phyla. For instance, the GM of children from México City showed a greater dominance of Firmicutes, Bacteroidetes, Actinobacteria, and Verrucomicrobia, whereas the GM of Me’phaa children was additionally represented by other phyla, including Tenericutes, Proteobacteria, Actinobacteria, Euryarchaeota, Spirochaete, Cyanobacteria, and Elusimicrobia. In addition, Me’phaa children not only had a greater abundance of these phyla, but also had other exclusive groups. The following eleven phyla were exclusively present in Me’phaa children, some with low abundances (under 0.01%); Chloroflexi, Deinococcus-Thermus, Acidobacteria, Fibrobacteres, Planctomycetes, Gemmatimonadetes, Latescibacteria, Nitrospirae, Lentisphaerae, Hydrogenedentes, and Aminicenantes (Figure 2C–H). Although our sample size from the urban population (*n* = 13) is a limitation to obtain conclusive results related to the most contrasting changes in GM between these two populations, previous studies carried out in Mexican children inhabiting cities resemble our results in terms of diversity, richness, and the absence of the groups exclusively found in Me’phaa communities [59,60].

A high diversity of GM taxa may be important because several low-abundance taxa are crucial for the homeostasis and maintenance of functions in human GM [58]. For instance, communities in Senegal [61] showed a high prevalence and diversity of Planctomycetes, which has antimicrobial activity [62]. Furthermore, metabolic reconstruction from Senegalese communities has suggested an anaerobic fermentative pathway and the ability to degrade multiple polysaccharides and glycoproteins from these exclusive groups [62,63,64]. We also identified a few taxa that, to our knowledge, have not been previously reported in human GM, i.e., Hydrogenedentes, Fibrobacteres, and Nitrospira. These taxa could contribute to different metabolic functions that have not previously been described in “non-Westernized” populations. For instance, the phylum Hydrogenedentes is a versatile carbon and energy-yielding chemotrophic metabolic group, associated with nitrogen, carbon, and sulfur pathways [65], whereas Fibrobacteres are primary degraders of cellulosic plant biomass in herbivore guts, which has prompted the suggestion that cellulose degradation may be a unifying feature of the phylum [66]. In children from México city we also found exclusive groups, but these were associated with Westernization, such as Saccharibacteria which has been associated with the degradation of sugar compounds [67] and there was a greater abundance of the Verrucomicrobia phylum, composed of environmental microorganisms and commonly abundant in the human gut after antibiotic exposure [24].

Another interesting result found in this comparison is related to the ratio of Firmicutes to Bacteroidetes abundance (F/B). This ratio has been used as a biomarker of several health conditions in Westernized societies and provides important information on the host’s lifestyle, diet, and metabolic function [68,69,70]. It has been proposed that an approximate F/B ratio of 3:1 or greater can be a predictor of childhood obesity [71]. Furthermore, a more specific taxonomic level (e.g., family) has shown that greater levels of Ruminococcaceae (i.e., Firmicutes) and depleted levels of Bacteroidaceae and Bacteroides (i.e., Bacteroidetes) are associated with obesity [69]. However, a high F/B ratio can also help extract more energy from fibers [18]. Interestingly, while Me’phaa children had F/B ratios similar to those reported in association with obesity in Westernized lifestyles, only two of the children (6% of the group) were obese (Appendix A). The Me’phaa lifestyle, characterized by low access to caloric food and a bacterial configuration that allows nutrients to be absorbed more efficiently would be expected to be beneficial in this context, without necessarily indicating excessive caloric intake. Nevertheless, the inference of functionality from a phylum level or even lower taxonomic levels is complicated due to the great diversity and complexity of the species, and this association needs further research in larger samples, using techniques that allow for higher taxonomic resolution and selecting bacteria with well-known physiological functions. Currently, we are just beginning to describe and understand the microbiota in traditional populations around the world, which is introducing new perspectives regarding what constitutes a “good” microbiome for human health.

Some species have been cataloged as emerging bioindicators of human health in these traditional communities. For instance, the family Christensenellaceae is a relatively recently described bacterial family [72]. This family is highly heritable, and its presence has been linked with the microbial ecology of the human gut and several diseases including obesity and inflammatory bowel disease [72]. Here, we found that the relative abundance of all species of this family was significantly higher in children from the Me’phaa community than from México City (Figure 4). A difference of this magnitude has not been reported before in communities from the same country with contrasting lifestyles [21]. This leads to new questions about whether or not this family provides some services to the health of these indigenous people, and if so, whether or not the lack of these taxa impact health in a Westernized lifestyle. One possibility is that this group was present in pre-Columbian populations but was lost in children from México City. In contrast, another possibility is that the group is replaced by others with similar functions, an event known as functional redundancy [2]. It would be revealing to carry out a meta-analysis comparison of our sequences with other non-Western communities such as the Bangladeshi or Malawi children datasets to answer these questions [22,50].

As expected, the genus *Akkermansia* was less abundant in children from México City than the Me’phaa children (Figure 4). This genus is related to the reduction of weight gain and fat accumulation, similar to what occurs with the Christensenellaceae family [72]. *Akkermansia* also improved glucose tolerance and reduced inflammation and metabolic endotoxemia in animal models of diabetes and obesity [73]. Interestingly, this genus also has a high capacity to degrade the mucus of the intestinal epithelium [73]. In a healthy state, the intestinal epithelium together with the mucus layer act as a physical barrier to bacteria and foreign antigens [74]. Therefore, at high abundances, *Akkermansia* can also have a potential role as a pathogen in diverse diseases (e.g., Alzheimer’s disease) [73]. In conclusion, even though Christensenellaceae and *Akkermansia* participate in similar inflammatory and metabolic functions, they have the potential to affect the host’s long-term health.

The high abundance of *Prevotella*, a VANISH group, among Me’phaa children is consistent with previous reports from other traditional communities, such as the Yanomami (Amerindians) and the BaAka (Africans) [48,75]. It has been suggested that *Prevotella* and *Treponema* are associated with high fiber intake, maximizing metabolic energy extraction from plant polysaccharides [18,54]. In addition to *Prevotella*, *Eubacterium* has also been associated with vegetarian diets within traditional communities [23,50,76] which resembles our results for the Me’phaa children. Similar to these results, the genera *Streptococcus* and *Clostridium sensu-stricto*, and members of the Erysipelotrichaceae family (Figure 5), were more abundant in the Me’phaa community.

Since the industrial revolution, there have been numerous diet and life practice changes in Westernized communities [77], which have led to decreasing microbial diversity. This decrease affects the microbial enzymatic capacity for degrading nutrients and many forms of complex polysaccharides in human diets [78,79]. Additionally, the depletion of microbial diversity can be associated with a broad range of inflammatory diseases, such as allergies, asthma, inflammatory bowel disease, obesity, and associated non-communicable diseases [12,80]. In this sense, although the study of non-Westernized populations is crucial to evaluate possible therapeutic properties of bacterial lineages that have been lost from Westernized human populations, most non-Westernized lifestyle communities are in decline [12,51]. Further studies also need to evaluate more than bacterial community composition, which on its own is not a reliable predictor of disease or bacterial functionality [2,22,81]. Hence, the contribution of specific taxa, their metabolic pathways, their interactions with other members of the GM and with their host is a new priority for microbiome research in these populations [12,80]. The use of OMICS approaches to study and compare non-Westernized populations has the potential to reveal functional consequences of these changes in the context of the microbial ecology of the gut and its impact on human evolution, biology, and health [82].

## Figures and Tables

**Figure 1 microorganisms-08-01592-f001:**
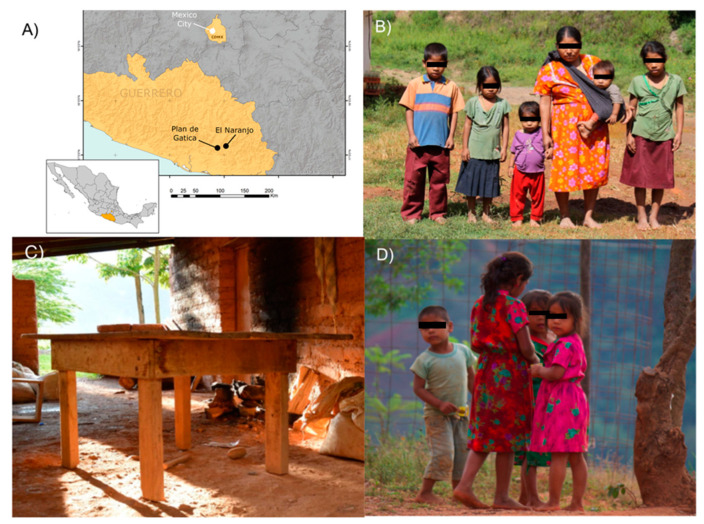
Maps of sample populations and lifestyle conditions in the Me’phaa population. (**A**) Sampling locations for this study. Map displaying the geographical locations taking as reference the south-central region of México in the state of Guerrero (black points) and México City (white point); (**B**) A representative Indigenous family from the Me’phaa community sampled; (**C**) The typical house construction observed in Me’phaa community; (**D**) Males and females children from 5 to 8 years old. Photos by I.G.-S.

**Figure 2 microorganisms-08-01592-f002:**
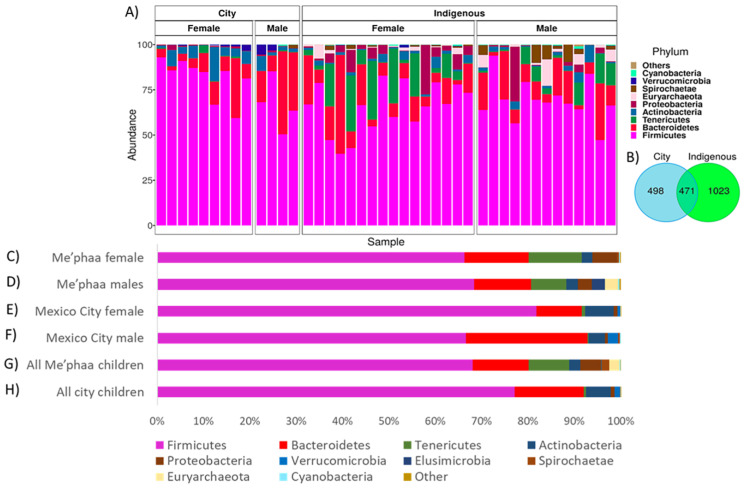
Relative abundance of microbiota. (**A**) Distribution of bacterial composition (16S rRNA V4) at phylum level of male and female children from México City and the Me’phaa community, phyla with relative abundances <1% were included in the “others” category; (**B**) Venn diagram of ASVs from México City and the Me’phaa community; (**C**) Relative abundance of gut microbiota (GM) female children from Me’phaa community; (**D**) Relative abundance of GM male children from the Me’phaa community; (**E**) Relative abundance of GM female children from México City; (**F**) Relative abundance of GM male children from the México City community; (**G**) Relative abundance of GM from all children in the Me’ phaa community; (**H**) Relative abundance of GM from all children in the México City community.

**Figure 3 microorganisms-08-01592-f003:**
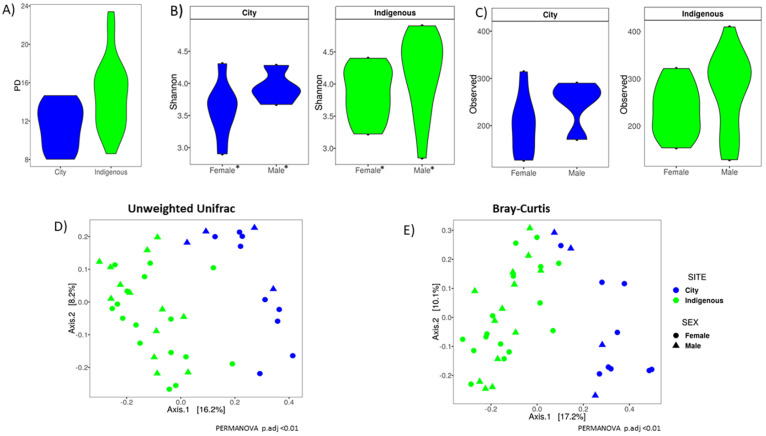
Alpha diversity indexes and Beta ordinations of children GM by population and sex. The violin plot with median of (**A**) Faith’s phylogenetic diversity (PD) index by population; (**B**) Shannon index; (**C**) Observed ASV’s. * corresponds to significant differences (*p* < 0.05); (**D**) Corresponds to Unweighted Unifrac; (**E**) Corresponds to the Bray–Curtis GM children analysis. For México City (City) and indigenous (Me´phaa) communities, blue and green were used, respectively. Differences were statistically significant in the PERMANOVA test at a level of *p* < 0.01 only for location, but not for the gender of the child in both distance metrics (**D**,**E**).

**Figure 4 microorganisms-08-01592-f004:**
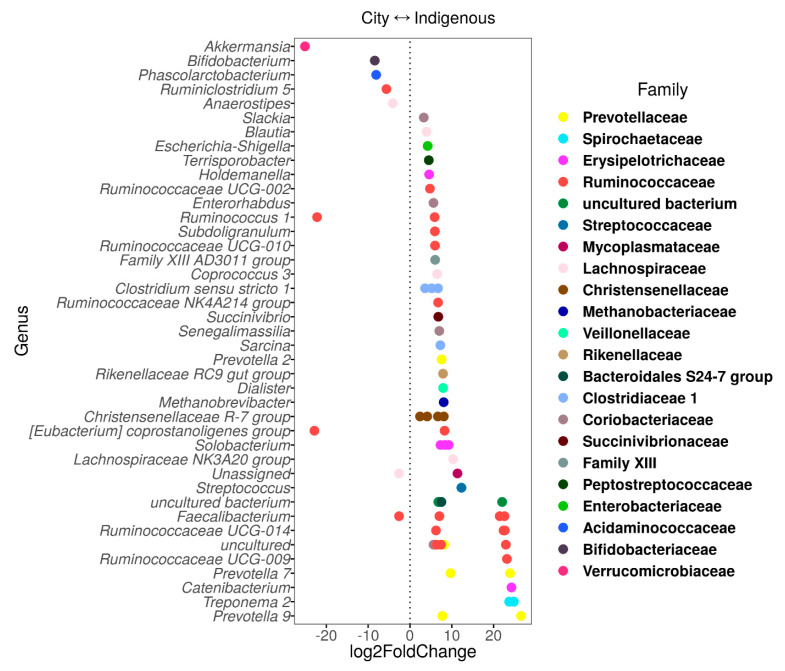
Log2 fold-change analysis of the combined children GM from México City (City) and indigenous (Me´phaa) communities. Fecal-prokaryotic ASVs grouped by genus and colored by family (legends). Log2 fold-change values indicate the strength and direction of the association to City (<0) and Me’phaa (>0) children. Genus observed in this figure were statistically significantly different between locations at *p* < 0.01 corrected with the FDR (false discovery rate) method.

**Figure 5 microorganisms-08-01592-f005:**
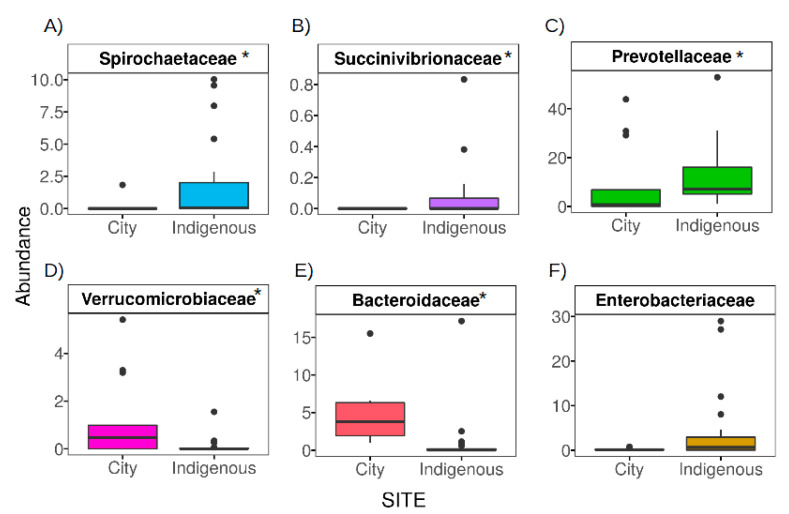
VANISH and BloSSUM GM boxplots from México City (City) and indigenous (Me´phaa) communities. VANISH and BIoSSUM boxplots show family taxa between México City and Me’phaa children’s fecal microbiota. The VANISH taxa are composed of Spirochaetaceae, Succinivibrionaceae, and Prevotellaceae families (**A**–**C**); the BIoSSUM family taxa are Verrucomicrobiaceae, Bacteroidaceae, and Enterobacteriaceae (**D**–**F**). * Corresponds to significant differences between communities (*p* < 0.01).

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
