# Peer review of "Gut Microbiome in Children from Indigenous and Urban Communities in México: Different Subsistence Models, Different Microbiomes"

_microorganisms, 2020, doi:10.3390/microorganisms8101592_

Round 1

Reviewer 1 Report

The authors investigated differences in the faecal microbiota between Mexico city children and rural children of an indigenous Mexican community. The study is interesting given the importance of life-style in the gut microbiota assembly and the fact that in westernised societies there is a loss of gut microbiota diversity correlating with disease. The manuscript will benefit from an extensive language revision in order to address not only grammatical issues but also low scientific accuracy in some sentences. The Discussion must be revised because in many parts it reads like a literature review. Some other issues must be addressed as indicated below.

Why were children chosen over adults? Why not adults too? The reasoning why you examined children is not clear. 

The manuscript language needs extensive revision: Lines 15-18: extensive rephrasing is needed in order to make the sentences more accurate and grammatically correct. Importantly, what is 'faecal microbiome approach in children'? This is very vague.

Line 36: it seems more accurate to mention peptides and other molecules instead of proteins unless you want to mention certain proteins.

Rephrasing is needed for accuracy of scientific meaning and correct use of terminology: lines 41-42: please rephrase 'networks of communication of GM'. Lines 44-45: 'one third...to most people'. Lines 46-47.

Lines 69-74: All this part needs rephrasing. E.g. rephrase for accuracy and grammar 'in particular among bacteria...societies'. My suggestion is to make simple phrases/brake down long phrases into smaller simpler ones that convey a clear message. What is the meaning of 'components of the host's fitness'. The whole part needs references too.

Lines 80-84, 105: please break into smaller sentences. Also, 'microbiota abundance' is not very accurate; is it genes, taxa, quantity or quality? Please rephrase.

Line 96: is there a reference?

Line 104: Is it stability or do you mean establishment and development? Please rephrase.

Line 106: 'opposed' is not used correctly, please rephrase with another verb.

Line 132-133: were there only 13 children from Mexico city? this seems like a low number to extract safe conclusions for the whole community. Even 29 is a low number. Please put a note down into the manuscript to recognise this.

Do you have any dietary data that you can present, correlate with your findings and discuss?

Line 153: why did you chose to amplify only V4 region? not for example v3-v4? is this routinely used in microbiota analysis? references?

Line 182: how did you generate the venn diagramme?

Line 202: I find the title not very representative of the text that follows: There is no deep taxonomic analysis in the following lines and it seems that this is not the main goal anyway. Please rephrase with a more representative title.

Lines 213-223: Could you add headings in Figure 2 so that it is clear what is being presented, especially for C-F. In panel A could you add an average of all F(city), all M(city) and the same for F and M of the indigenous communities? Since you discuss average in the text it would be better to show it in the Figure too. Why isn't there any Figures of Family level analysis? Please add.

Line 217-219: please put notes in the parenthesis what are these number bcs it doesn't go without saying that you refer to F and M.

Line 219-220: you refer to diversity but without directing the readership to a certain Figure. Please address.

Line 234: A 'comparison of GM' is very vague. Please rephrase capturing the most significant findings to be described.

Line 249: If this is a padj value for multiple testing please mention it.

Figure 3: Please add y-axis scale to all sub-Figures. A demonstration of the direct comparison between City vs Rural of the average alpha diversity indexes should be shown in one Figure with stats. Here it's mostly differences between gender shown. Figure 3 should be redesigned. 

Figure 4: Please redesign the Figure with stats on it and with head titles so that it is clear with one look what is shown. Did you do Weighted too? Why not presented? 

Line 258: 'Differences...' when testing for site/location only or and/or gender? Please specify were the PERMANOVA refers to.

Please use italics for Gender and Species throughout the text.

Lines 271-272: how is significance denoted in the Figure or explained in the legend?

Line 281: Please add a short note explaining the selection.

Line 282: Did you examine infant too? Please correct if you mean the younger children in your examined cohort.

Line 274: Please rephrase with a title to capture your most significant findings described in the section to follow.

Line 295: Are these padj values for multiple corrections?

Line 303, 306: Please keep richness or diversity depending on where you refer to; 'abundance' would refer to the abundance of taxa.

Line 310: reference?

Lines 318-320: Please rephrase for accuracy.

Lines 353-354: Please rephrase, for scientific accuracy and grammar. 

Line 366368: Please rephrase for accuracy and grammar; break into two sentences if needed.

Line 394: reference please.

The Discussion is too long and some parts read more like a literature review rather than a focused-on-results discussion e.g. 392-400; 422-450. This must be addressed and the Discussion should be more succinct, focused and significantly shortened. In the Discussion you haven't discussed much the children life-style trying to match with your results. 

Apologies if I missed it, but where in the main text do you refer to supplementary materials?

Author Response

 Why were children chosen over adults? Why not adults too? The reasoning why you examined children is not clear.

Answer: We agree with your suggestion. We have included a couple of sentences reinforcing the reasons for using children instead of adults. For example, “Childhood is an important age for the establishment and development of GM”. Please see lines: 132-133

Also, we are including the few information we have about GM in children older than 2 years old in comparison to the extensive studies done in Adults and infants from 0-2 years old. In addition to the corresponding reference. Please see lines:  77-81

The manuscript language needs extensive revision: Lines 15-18: extensive rephrasing is needed in order to make the sentences more accurate and grammatically correct. Importantly, what is 'faecal microbiome approach in children'? This is very vague.

Answer: Thank you for your comment. We rephrased lines 15-18 and the sentence was adjusted  to:  “Using children fecal samples for 16rDNA characterization, a comparison of two Mexican communities with contrasting lifestyles: “westernized” (Mexico City) and “non-westernized” (Me’phaa indigenous group) was evaluated”. Please see lines:19-24

Line 36: it seems more accurate to mention peptides and other molecules instead of proteins unless you want to mention certain proteins.

Answer: Thank you. Yes, we agree with your comment and we changed peptides instead of proteins (Line 44).

Rephrasing is needed for accuracy of scientific meaning and correct use of terminology: lines 41-42: please rephrase 'networks of communication of GM'. Lines 44-45: 'one third...to most people'. Lines 46-47.

Answer: Thank you for the suggestion. We have rephrased lines 41-47. Also, we modified 'networks of communication of GM' by the term “GM connectivity”. We used this term in accordance with our last recent publication (Carrillo-Ramirez et al., 2020). Here, we implemented a complex systems approach based on Network Theory methods (one of the main streams in complexity). This approach has shown to be useful to study ecosystem dynamics, and it is a complementary tool to the traditional methodologies that evaluate composition and abundance in an ecosystem (i.e. alpha diversity index). See lines: 50-51; 54-58

Lines 69-74: All this part needs rephrasing. E.g. rephrase for accuracy and grammar 'in particular among bacteria...societies'. My suggestion is to make simple phrases/brake down long phrases into smaller simpler ones that convey a clear message. What is the meaning of 'components of the host's fitness'. The whole part needs references too

Answer: Thank you for your suggestion. We rephrased the sentence and simplified the paragraph. We also rephrased “components of the” host´s fitness” and added references [Jha et al 2018]. Hopefully, you find a substantial improvement in this new paragraph. Please see lines: 94-107

Lines 80-84, 105: please break into smaller sentences. Also, 'microbiota abundance' is not very accurate; is it genes, taxa, quantity or quality? Please rephrase.

Answer: Thank you. We agree with you and we shortened the phrases into smaller ones. We hope you find this new paragraph improved. Also, we modified 'microbiota abundance' for 'microbiota richness'

Line 96: is there a reference?
Answer: Thank you for your comment. This information was provided by the community. Therefore, no reference was contemplated. However, we have extensive work in these communities for approximately 8 years (see Leóngomez et al., 2020; Ramirez-Carrillo et al., 2020; Borda-Niño et al., 2016; Camacho, 2007) in collaboration with the Non-Governmental Organization Xuajin Me´phaa. This work allowed us to obtain these data since several socio-demographic aspects of these people were evaluated. These data are in preparation for publication. Moreover, for the present study we only obtain fecal samples from individuals that meet the requirements mentioned in this paragraph.

Line 104: Is it stability or do you mean establishment and development? Please rephrase.

Answer: Thank you for your observation. We rephrased the sentence to “Therefore, the aim of the present study is to explore whether these contrasting lifestyles influence the ecosystemic dynamics of GM in childhood, an important age for the establishment and development of GM”. Please see lines:131-133

Line 106: 'opposed' is not used correctly, please rephrase with another verb.

Answer: We agree with your comment. We changed “opposed” for “contrasting” (Line 136).

Line 132-133: were there only 13 children from Mexico city? this seems like a low number to extract safe conclusions for the whole community. Even 29 is a low number. Please put a note down into the manuscript to recognise this.

Answer: We agree with your comment. We added several sentences referring to this issue. Please see lines: in methods and Lines 167-169: in discussion: 416-421

Do you have any dietary data that you can present, correlate with your findings and discuss?

Answer: Yes, we do have dietary data from these children. It would be very useful to include them to discuss our results. However, these data were used in an additional manuscript recently submitted (Please see; Ramirez-Carrillo et al., 2020; DOI: 10.1101/2020.07.25.221408). Hence, it was not possible to include them here. Furthermore, the main goal of this manuscript was to show changes in GM due to general patterns in lifestyle practices occurring in these contrasting populations, such as medication, access to health services, and diet in general. In this case, we did not intend to assess how the amount of particular food intake affects GM. Again, we fully agree that this information would be very important, and we look forward to it being released in the future.

Line 153: why did you chose to amplify only V4 region? not for example v3-v4? is this routinely used in microbiota analysis? References?

Answer: We agree with you that it is more common to use V1-V3 or V3-V5. However, according to Allaband et al. 2019, the V4 primers are also valuable because of the capability of picking up archaea (p.e. Methanobrevibacter, among others) which are also important for the gut.

Line 182: how did you generate the venn diagramme?

Answer: The Venn diagram was plotted with the ASVs by populations using the vennDiagram function of the limma library (Ritchie 2015). Please see lines: 220-221

Line 202: I find the title not very representative of the text that follows: There is no deep taxonomic analysis in the following lines and it seems that this is not the main goal anyway. Please rephrase with a more representative title.

Answer: Thank you. Yes, we agree with your suggestion and changed the title for “ Microbial taxonomic comparison from Mexico City and Me'phaa communities”.See lines: 258-259

Lines 213-223: Could you add headings in Figure 2 so that it is clear what is being presented, especially for C-F. In panel A could you add an average of all F(city), all M(city) and the same for F and M of the indigenous communities? Since you discuss average in the text it would be better to show it in the Figure too. Why isn't there any Figures of Family level analysis? Please add.

Answer: Thank you for the observation. We added the headings to Figure 2. We also added a supplementary figure at Family level with relative abundances. However, figures C-F corresponds to the relative abundance from the average of each F(city), M(city), F(indigenous) and M(indigenous). In fact, we wanted to illustrate the variance per sample as it is shown by   Figure 2A and for that reason, we consider that adding another plot with the averages is not crucial.

Line 217-219: please put notes in the parenthesis what are these number bcs it doesn't go without saying that you refer to F and M.

Answer: Thank you for the suggestion. We added brief notes for referring to Tenericutes and Proteobacteria for the City females and males. Please find in lines: 279-280

Line 219-220: you refer to diversity but without directing the readership to a certain Figure. Please address.

Answer: Thank you for your comment. We addressed the corresponding figure for line: 285

Line 234: A 'comparison of GM' is very vague. Please rephrase capturing the most significant findings to be described.

Answer: Yes. we totally agree, we changed 'comparison of GM' for “Alpha diversity, clustering ordination and Family/Genus comparison of GM in children from Mexico City and Me'phaa communities”. See lines 300-301

Line 249: If this is a padj value for multiple testing please mention it.

Answer: Many thanks for your assertive comment. Yes, this is a p-value adjusted for multiple comparisons. In our case, we adjusted p values using the holm´s method. This method provides us a balance between errors type 1 and 2 since it is less conservative than Bonferroni, but more than others such as Hochberg or Hommel (Holm, 1979). A simple sequentially rejective multiple test procedure. Scandinavian Journal of Statistics, 6, 65–70.)

We added this information in lines:246-248.

Figure 3: Please add y-axis scale to all sub-Figures. A demonstration of the direct comparison between City vs Rural of the average alpha diversity indexes should be shown in one Figure with stats. Here it's mostly differences between gender shown. Figure 3 should be redesigned.

Answer: Thank you for the suggestion. We incorporated y-axis for the missing plots. On the other hand, the reason for presenting figure 3 in this way is because the Faith PD was the only one that was statistically significant between populations independently of sex (Please see lines 303-305), while Shannon and observed ASV´s were different between sexes, but only a tendency between communities. In both populations, females presented less alpha diversity than males (Please see lines 309-313). Moreover, indigenous children, both men, and women showed more alpha diversity than the same sex in the city. Considering your opinion, we redesigned this figure. Nevertheless, if you consider that it needs other substantial change by grouping both sexes and placing only the populations, please let us know. Finally, we included Stats in figure legend.

Figure 4: Please redesign the Figure with stats on it and with head titles so that it is clear with one look what is shown. Did you do Weighted too? Why not presented?

Answer: Thank you for this comment. We redesigned the ordination plots with head titles and stats were included in the figure legend. In addition, we also included in Figure Sup. 3 The PcoA of Weighted Unifrac distances. The corresponding statistical information was also included in lines: 321-324.

Line 258: 'Differences...' when testing for site/location only or and/or gender? Please specify were the PERMANOVA refers to.

Answer: Thanks for this observation. We only find differences in Location in both distance metrics. We included this information in lines: 320-326

Please use italics for Gender and Species throughout the text.

Answer: Thank you for your observation. We changed all Gender and Species in italics throughout the text.

Lines 271-272: how is significance denoted in the Figure or explained in the legend?

Thanks for your observation. All genus showed in Figure 5 were statistically significantly different between locations at p< 0.01. The p-value was adjusted with the FDR (false discovery rate) method. We modified this sentence for greater clarity.  See lines 344-346

Line 281: Please add a short note explaining the selection.
Answer:Thank you for the observation. We rephrased the paragraph to “According to Sonnenburg and Sonnenbgurg (2019), there are differences in the abundance of VANISH taxa (volatile and/or associated negatively with industrialized societies of humans) and those positively associated with societies of urbanization/modernization (BIoSSUM).For this manuscript, we considered community and sex of the children as predictors of the following families: Bacteroidaceae, Enterobacteriaceae, Verrucomicrobiaceae, Prevotellaceae, Spirochaetaceae and Succinivibrionaceae (Fig. 6).” Hopefully, it is clearer in this new version. See lines: 352-357

Line 282: Did you examine infant too? Please correct if you mean the younger children in your examined cohort.
Answer: We agree with your comment. We replaced infant for children (Line 365).

Line 274: Please rephrase with a title to capture your most significant findings described in the section to follow.

Answer: Thank you for the suggestion. We changed the title to “The contrasting relationship between VANISH and BloSSUM groups between indigenous and city children”. Lines:349-350

Line 295: Are these padj values for multiple corrections?

Answer: Thanks for your observation. Yes, we showed p.adj values using holm´s method. We included this information in lines :246-248.

Line 303, 306: Please keep richness or diversity depending on where you refer to; 'abundance' would refer to the abundance of taxa.

Answer: Thank you for the correction. We changed “abundance” for richness

Line 310: reference?

Answer: Thank you for the comment. We decided to remove this sentence.

Lines 318-320: Please rephrase for accuracy.

Answer: Thanks for your observation. We rephrased lines 403-404

Lines 353-354: Please rephrase, for scientific accuracy and grammar.

Answer: Thanks for your observation. We rephrased the paragraph. Please find in lines: 440-444

Line 366-368: Please rephrase for accuracy and grammar; break into two sentences if needed.

Answer: Thank you for your comments. We rephrased the sentence hoping to improve to text. The new version corresponds to lines:456-461.

Line 394: reference please.
Answer: Thank you. We decided to remove the sentence

The Discussion is too long and some parts read more like a literature review rather than a focused-on-results discussion e.g. 392-400; 422-450. This must be addressed and the Discussion should be more succinct, focused and significantly shortened. In the Discussion you haven't discussed much the children life-style trying to match with your results.

Answer: Thank you. We shortened the discussion. I hope you find this version shorter and more related to the specific discussion of our results.

Apologies if I missed it, but where in the main text do you refer to supplementary materials?

Answer: You were right, thank you, we added the supplementary materials references in the text. (Lines 220, 286).

Reference:

Leongoméz, J. D., Sánchez, O. R., Vásquez-Amézquita, M., Valderrama, E., Castellanos-Chacón, A., Morales-Sánchez, L., ... & González-Santoyo, I. (2020). Self-reported health is related to body Height and waist circumference in rural indigenous and urbanised Latin-American populations. Scientific reports10(1), 1-13.

Ramírez-Carrillo, E., Gaona, O., Nieto, J., Sánchez-Quinto, A., Cerqueda-García, D., Falcón, L. I., ... & González-Santoyo, I. (2020). Disturbance in human gut microbiota networks by parasites and its implications in the incidence of depression. Scientific Reports10(1), 1-12.

  1. Borda-Niño, M. Carranza, S. Diego Hernández-Muciño, and M. Muciño-Muciño, “Restauración productiva en la práctica: el caso de las comunidades indígenas me’phaa de la Montaña de Guerrero”, Perspectivas sociales en América Latina y el Caribe. 2016, Buenos Aires: Vázquez Manzzini Editores.

Camacho, “Montaña de Guerrero pobreza y militarización | Revista Contralínea” 2007, Available online: https://www.contralinea.com.mx/archivo/2007/enero/htm/montana_guerrero_militares.html

Reviewer 2 Report

Sánches-Quinto et al perform an assessment of fecal samples using a 16S rRNA V4 protocol to compare two different populations of children in Mexico. One of the populations is “westernized” the other is “non-western”. There are a variety of dietary and lifestyle differences between these groups. Some of the primary finds correspond to existing microbiome work in “non-western” populations including differences in alpha diversity and putative enrichment for organisms like Trepomena.    Better understanding how different human populations is of incredible importance for the microbiome field, and these samples will likely receive a high amount of reuse as has been seen in many other sample sets. However, the analyses included here are a bit limited. Specifically, there is limited focus on the use of compositionally aware tools (e.g., https://www.nature.com/articles/s41467-019-10656-5), and I think there is a huge opportunity here for insight through meta-analysis with other populations. A few possible studies are noted below that might be interesting to considered, although the authors listed a few additional and relevant studies in the discussion.   Last, from a reuse perspective, the authors may want to consider including their data in Qiita (https://qiita.ucsd.edu). This will simplify the reuse of their data, and draw attention to these important sample collections.   45: that seems in contrast to the findings of the human microbiome project were a core microbiome at the OTU level was not observed? 112: was this done under the approval of an IRB?  Fig 1, these images seem identifying — has informed consent been obtained that allows for including pictures of these individuals? 204-223: these types of analyses are prone to a high degree of error as they do not factor in a reference frame. Would the authors consider using compositionally aware methods like Songbird? 234: it would be really interesting to perform a meta-analysis to see how (a) these data relate to Yatsunenko et al 2012 and (b) Cuesta-Zuluaga et al 2018. Both should be technically compatible with the protocol.  281: sonnengurg -> sonnenburg? Figure 6, is there any chance of increasing the specificity? Family level is rather coarse 357: F/B hasn’t been shown to have any actual support… these groups are separated by 100s of millions or more years of evolution. I agree that log ratios should be used, however ideally they are objectively picked and at high resolution. 368: it may be interesting to perform a meta-analysis with microbiome samples from Bangladeshi children (https://science.sciencemag.org/content/365/6449/eaau4732) or Malawi (https://www.ncbi.nlm.nih.gov/pmc/articles/PMC3667500/)? Food insecurity is an important issue for these populations as well, and it would be interesting to see if there are signals in common among the datasets.  409: Again, meta-analysis would be cool here — are they the same ASVs? 435: “is likely” is overstating the evidence… Discussion section: many of these paragraphs are massive. Readers would benefit if they were broken up further.

Author Response

Sánchez-Quinto et al perform an assessment of fecal samples using a 16S rRNA V4 protocol to compare two different populations of children in Mexico. One of the populations is “westernized” the other is “non-western”. There are a variety of dietary and lifestyle differences between these groups. Some of the primary finds correspond to existing microbiome work in “non-western” populations including differences in alpha diversity and putative enrichment for organisms like Trepomena.    Better understanding how different human populations is of incredible importance for the microbiome field, and these samples will likely receive a high amount of reuse as has been seen in many other sample sets.

However, the analyses included here are a bit limited. Specifically, there is limited focus on the use of compositionally aware tools (e.g., https://www.nature.com/articles/s41467-019-10656-5), and I think there is a huge opportunity here for insight through meta-analysis with other populations. A few possible studies are noted below that might be interesting to considered, although the authors listed a few additional and relevant studies in the discussion.   Last, from a reuse perspective, the authors may want to consider including their data in Qiita (https://qiita.ucsd.edu). This will simplify the reuse of their data, and draw attention to these important sample collections.

Answer:

We appreciate your constructive suggestions; we totally agree that these suggestions will be very useful to improve our manuscript’s quality. Therefore, we revised our manuscript according to your advice one by one.

We agree with the reviewer 2 that it would be great to compare our set with other previously reported sequences. Nevertheless, the main goal of this study was to compare two of the most contrasting communities within Mexico, a country with a great diversity in terms of  lifestyle practices (and scarce information published regarding this kind of comparisons).We appreciate all of your contributions and we will continue working to obtain this comparison that you recommend for a novel study goal.

In this sense, we included this recommendation as a perspective (Please see lines:478-480).

Regarding Qiita, we agree that the data should be used for further knowledge, comparisons and future analysis. Therefore, all raw-data are available in the bioproject PRJNA593240 in the NCBI platform for any further reuse.  

 45: that seems in contrast to the findings of the human microbiome project were a core microbiome at the OTU level was not observed?

Answer: We appreciate this observation, effectively the Human microbiome project was not conclusive about a core microbiome at the OTU level. Nevertheless, by focusing on the gut microbiota Falony et al., 2016, and then confirmed by Rojo et al., 2017 revealed that there are 14 genera that may be considered as the core gut microbiota. In consequence, we now modified this sentence referring specifically to Gut microbiota at genus level. Please see lines: 54-58.

112: was this done under the approval of an IRB?  Fig 1, these images seem identifying — has informed consent been obtained that allows for including pictures of these individuals?

Answer: We appreciate your observation. All photographs have individuals or parents’ approval for this manuscript. We now included this information in the Methods section. In addition, we also contain an IRB approval for this project by the National Autonomous University of Mexico Committee of Research Ethics (FPSI/CE/01/2016). This information is now included in lines: 250-256.

204-223: these types of analyses are prone to a high degree of error as they do not factor in a reference frame. Would the authors consider using compositionally aware methods like Songbird?

Answer: We appreciate this observation and we agree. Nevertheless, the section mentioned by the reviewer just describes the overall composition. The analysis to detect differences between populations was performed with methods to handle compositional data as deseq2, which detects the fold changes of features just like songbird does, and Zero Inflated regression models. Results of these analyses are present below in the manuscript.

234: it would be really interesting to perform a meta-analysis to see how (a) these data relate to Yatsunenko et al 2012 and (b) Cuesta-Zuluaga et al 2018. Both should be technically compatible with the protocol.

Answer: Yes! Definitively! We agree with your comment. However, our objective for this specific research was to compare two Mexican communities with contrasting lifestyles. Therefore, we consider that adding other world-wide non-westernized communities could divert the focus of attention on the communities that were originally chosen as targets.

 281: sonnengurg -> sonnenburg?

Answer: Thank you for the observation. We changed the text to “These groups were selected according to Sonnenburg and Sonnenburg (2019)”. Line:352

 Figure 6, is there any chance of increasing the specificity? Family level is rather coarse

Answer: Thank you for the observation. We agree that Family level could be coarse.  However, we were interested in referring to the VANISH and BloSSUM groups as previous authors have reported recently, who used families at the comparative taxonomic level (Sonnenburg and Sonnenburg, 2019). This allowed us to compare our Mexican communities with those reported by mentioned authors, which have suggested that these families are the most influenced by contrasting lifestyle practices between westernized and non-westernized. Nevertheless, we also obtained more specificity for our two communities, implementing the log2 fold change analysis showed in figure 5. Here, we also found that most of the ASVs contrasting between communities belonged to these VANISH and BIoSSUM families previously reported.  

357: F/B hasn’t been shown to have any actual support… these groups are separated by 100s of millions or more years of evolution. I agree that log ratios should be used, however ideally they are objectively picked and at high resolution.

Response: We absolutely agree with this timely comment. The F/B approximation has been recently questioned. In fact, we mentioned some of the limitations to associate the F/B ratio with a determined health status and more specifically to consider it as a hallmark of obesity. Nevertheless, considering this important suggestion, we reorder and change this section including the imperative need to use a higher resolution in these approximations, based on the physiological functions of the picked groups. Please see lines:440-454

368: it may be interesting to perform a meta-analysis with microbiome samples from Bangladeshi children (https://science.sciencemag.org/content/365/6449/eaau4732) or Malawi (https://www.ncbi.nlm.nih.gov/pmc/articles/PMC3667500/)?

Answer: Thank you for the suggestion. As we mentioned before, we think it would be exciting to make a comparison of our samples with other worldwide communities. However, the objective of this specific project is focused on Mexican communities. Although, we agree with you, and it is something that should be done in the near future. We included this prospect in lines 478-480 

Food insecurity is an important issue for these populations as well, and it would be interesting to see if there are signals in common among the datasets.  

Answer: Thank you. Yes, we do have dietary data from these children. It would be very useful to include them to discuss our results. However, these data were used in an additional manuscript recently submitted (Please see; Ramirez-Carrillo et al., 2020; DOI: 10.1101/2020.07.25.221408). Hence, it was not possible to include them here. Furthermore, the main goal of this manuscript was to present changes in GM predicted by the general patterns in lifestyle practices occurring in these contrasting populations, such as medication, access to health services, and diet in general. In this case, we did not intend to assess how the amount of particular food intake affects GM. Again, we fully agree that this information would be very important and we look forward to it being released in the future.

409: Again, meta-analysis would be cool here — are they the same ASVs?

Answer: Yes! It would be amazing! However, we believe that this comparison should be the next step for a possible next manuscript.

 435: “is likely” is overstating the evidence…

Answer: Thank you for your observation. We decided to change “is likely” for “could be associated” to avoid exaggerating the information

Discussion section: many of these paragraphs are massive. Readers would benefit if they were broken up further.

Thank you for the observation, we have synthesized the discussion into shorter paragraphs. We hope you find the new discussion more accurate.

Round 2

Reviewer 1 Report

The manuscript has been improved; some further important issues are listed below:

Language and style still need some improvement:

e.g. lines 16-18: 'important host's component' ... (GM).' needs rephrasing and to be split in two. 

Line 18: Rephrase: please use faecal samples from children and 16rDNA doesn't mean anything. Please write correctly: 16S rRNA gene 

Lines 18-20 for syntax: '...a comparison cannot be evaluated' please rephrase.

Line 41: please use GM dynamics instead of connectivity

Line 44, 46, 66: 'a sort of', 'etc', 'on the other hand' are not very formal, please erase.

Lines 46-47: which are the remaining groups? this does not make much sense. Please rephrase in a simple clear way lines 46-48 with reference. 

Reference is needed in line 64.

Lines 63-66: Please rephrase in a clear way to convey an exact message with references.

Lines 66-68: reference is needed.

Line 71: 'evaluating' is very vague; please specify.

Line 76: 'host's fitness' for what? please rephrase/specify.

Lines 77-87: extensive language and syntax revising is needed for clarity. More references are needed e.g. 83-85.

Please correct wherever needed e.g. 111: 'grow in a ... life-style' is not grammatically correct.

Lines 134-135: Dear authors, stating that 'it is important' does not necessarily make the statement important. Please just acknowledge the fact of low numbers and importantly the potential limitations maybe using references with similar numbers, without offering a personal opinion. 

Line 150: Please use a reference for why you used just the V4 for the community characterisation.

Line 174: what does '...by population' mean? Where was this plotted? R? Specify.

Please note in figures that p is adjusted.

Figure 2: 'agglomarated' is not correct in this case; please rephrase. 'Relative abundance of ... children' would refer to the abundance of children not their microbiota composition. Rephrase the whole legend for accuracy. Please add a column of the combined result in City and in Indigenous bcs further down you combine anyway (in the Supplementary figure too); the Venn diagramme takes into consideration all from each location and further down in Figure 3 you combine f&m. Please make C, D, E, F into bars next to each other for direct comparison and if you wish add the numbers (%) too. Further down in Discussion you mention a lot to these Phylum results (309-321) and yet there are no stats on the Phylum results. I think if you are to base so much of your discussion on Phylum you should do stats on these results.

Figure 3: The title of this figure and Figure 2 must be rephrased. 'Alpha diversity'is not enough to describe the figure. Also, why A) is so much bigger than B) and C) ? 

Line 237-239: You state ' The difference...., but not between...' Since you take the whole city (F&M) vs indigenous (F&M) for PD diversity, why don't you show the cumulative diversity in the Figure too rather than keeping m&f separately in A) ? The combined PD should be shown bcs this is what you discuss in the text and then for informative reasons, split into f and m. The whole Figure must be redrawn and for comparisons between the two locations you have to put them in the same box not in separate. Separate boxes would tell the readership that they refer to within-location comparisons. 

Line 237: No need to mention Faith's Index all the time.

Line 252: what doe you mean '... but a tendency' ?

Figure 4: Please make the title more accurate. Please add PERMANOVA results on the Figure. What we see here is the effect of location?

Figure 5: Please make the title more specific. Please specify that here you consider combined M&F from each location. Is that right? 

Figure 6: Specify title. Does the * next to the Family name in each plot show the initial test result or the post-hoc result? 

Lines 309-310: Dear authors, how is it that the fact that the bacteria taxa you mention to be 'common phyla reported for human diversity' is 'interesting' since this is a matter of fact ? Please rephrase bcs the whole sentence message is lost. 

Line 311-312, 312-313: Please specify to which communities you refer to to avoid misconceptions. Please rephrase 'may tell us about...'. Please rephrase the whole 312-313 bcs it does not convey any clear scientific message.

Lines 309-325: Please see comments on Figure 2.

Lines 367-369: Reference please.

Lines 377-387: Please rephrase for clarity of take-home message and add a reference in line 379.

Lines 394-395: Which are these? Please specify somehow bcs as it is now it reads as if Streptococcus, Clostridium snsu stricto and Erysipelotrichia are not commonly present, which is totally erroneous. Also, how is this intro statement binds/leads to what you discuss further down? Major revision here please (394-401).

Lines 399-406, 408: References are needed. 

Author Response

The manuscript has been improved; some further important issues are listed below:

Language and style still need some improvement:

Dear Reviewer:

The manuscript has been reviewed by a native English speaker. In addition, we have attended to all your comments.

We believe that the manuscript has been improved and hopefully, you will find this new version more suitable. 

e.g. lines 16-18: 'important host's component' ... (GM).' needs rephrasing and to be split in two. 

Answer: Thank you for the observation. We have rephrased lines 16-18. The new sentence can be found in lines 20-22

Line 18: Rephrase: please use faecal samples from children and 16rDNA doesn't mean anything. Please write correctly: 16S rRNA gene

Answer:We agree with your comment. We have changed “fecal” for “faecal” and “16S rDNA” for 16S rRNA gene respectively (lines 23-26).

Lines 18-20 for syntax: '...a comparison cannot be evaluated' please rephrase.

Answer: Thank you for the correction. We have changed this sentence, Please see lines 23-26

Line 41: please use GM dynamics instead of connectivity

Answer: Thank you for the suggestion. We have changed connectivity for dynamics. Line 53

Line 44, 46, 66: 'a sort of', 'etc', 'on the other hand' are not very formal, please erase.

Answer: We agree. We have deleted: 'a sort of', 'etc', and 'on the other hand'. Please fin the new sentence in line 55-58 and 91-92 respectively.

Lines 46-47: which are the remaining groups? this does not make much sense. Please rephrase in a simple clear way lines 46-48 with reference. 

Answer: Thank you for the observation. We have rephrased lines 46-48. The new sentence is in lines 59-63.: “Nevertheless, more than 2000 genera have been characterized to date in the human GM, and their abundance and composition can vary greatly among individuals, resulting in up to 80% genetic dissimilarity of the GM from one person to another [2], [10]”.

Reference is needed in line 64.

Thank you for the observation. 

We supported this information from Agnas et al., 2011 (i.e. reference 18 in the manuscript). These authors argued that: " Although the intestinal microbiota has been cataloged extensively in adults, and many recent reports were published for children between 0 and 2 years of age (Hopkins et al., 2005; Palmer et al., 2007; Bjorkstrom et al., 2009), few studies were conducted with the gut microbiota of older children (Hopkins et al., 2001; Chernukha et al., 2005; Enck et al., 2009; Paliy et al., 2009). ".  

We are aware that this reference is 9 years old and currently there are many more studies of these ages. However, the number of studies in children of these ages belonging to non-Western populations continues to be very scarce. Among other authors, this statement is supported by De Filipo in 2017, where it is mentioned that the exploration of GM at these ages is very scarce in this type of population. This quote is now included in our manuscript, please see lines:

Reference:

De Filippo, C., Di Paola, M., Ramazzotti, M., Albanese, D., Pieraccini, G., Banci, E., ... & Lionetti, P. (2017). Diet, environments, and gut microbiota. A preliminary investigation in children living in rural and urban Burkina Faso and Italy. Frontiers in microbiology, 8, 1979.

Hopkins MJ Macfarlane GT Furrie E Fite A Macfarlane S (2005) Characterisation of intestinal bacteria in infant stools using real-time PCR and northern hybridisation analyses. FEMS Microbiol Ecol54: 77–85.

Palmer C Bik EM Digiulio DB Relman DA Brown PO (2007) Development of the human infant intestinal microbiota. PLoS Biol5: e177.

Eckburg PB Bik EM Bernstein CN et al. . (2005) Diversity of the human intestinal microbial flora. Science308: 1635–1638.

Bjorkstrom MV Hall L Soderlund S Hakansson EG Hakansson S Domellof M (2009) Intestinal flora in very low-birth weight infants. Acta Paediatr98: 1762–1767.

Hopkins MJ Sharp R Macfarlane GT (2001) Age and disease related changes in intestinal bacterial populations assessed by cell culture, 16S rRNA abundance, and community cellular fatty acid profiles. Gut48: 198–205.

Paliy O Kenche H Abernathy F Michail S (2009) High-throughput quantitative analysis of the human intestinal microbiota with a phylogenetic microarray. Appl Environ Microb75: 3572–3579.

Lines 63-66: Please rephrase in a clear way to convey an exact message with references.

Answer: Thank you for the suggestion. We have rephrased lines 63-66. Hopefully you will find an improvement in the sentence.

Lines 66-68: reference is needed.

Thank you very much for this important observation. Now, we have been specific that this very small number of studies in Non-Westernized populations corresponds to those that have considered children in the age range that we use in our manuscript. This is supported by Ayeni et al., 2018 (reference 20)  and De Filippo et al., 2017 (now cited in the manuscript), among others. Please see changes in lines:90-95

Line 71: 'evaluating' is very vague; please specify.

Answer: We agree with your suggestion. We have replaced “evaluating” with “characterized” (Line 93)

Line 76: 'host's fitness' for what? please rephrase/specify.

Thank you very much for this important observation. We agree that the term host fitness is vague in this sentence. Therefore, we have changed this sentence specifying how changes in lifestyle can influence health and survival. This connects the information offered on the line XX, where we explain much of the role of GM in human health. Please see these changes on the line:97-101

Lines 77-87: extensive language and syntax revising is needed for clarity. More references are needed e.g. 83-85.

Totally agree with this useful suggestion. Therefore, we have completely modified this paragraph to give greater coherence to the text. We also included more references that support the paragraph written. Please see lines.114-128

Please correct wherever needed e.g. 111: 'grow in a ... life-style' is not grammatically correct.

Answer:Thank you. We have changed the sentence. Please see lines 152-157

Lines 134-135: Dear authors, stating that 'it is important' does not necessarily make the statement important. Please just acknowledge the fact of low numbers and importantly the potential limitations maybe using references with similar numbers, without offering a personal opinion. 

Answer: Thank you, we have changed the paragraph. The new sentence is: “Despite the limitation of not being a vast number of samples, other authors have previously used this approximation”.  Please see lines 186-187

Line 150: Please use a reference for why you used just the V4 for the community characterisation.

Answer: We appreciate your comment. However, we already included Allaband et al. 2019 (reference # 32) where they justify the use of V4 by obtaining Archaea organisms which are important in the gut microbiota.

Other important projects such as Yatsunenko et al 2012 have used this approximation.Please see lines 208-209

Line 174: what does '...by population' mean? Where was this plotted? R? Specify.

Answer:Thank you. We have rephrased the sentence to: “ A Venn diagram was plotted with the ASVs by location using the vennDiagram function of the limma library in R”. Hopefully you will find an improvement in this new sentence.Please see lines 234-235

Please note in figures that p is adjusted.

Answer: Yes, thank you. We have now added that p is adjusted.

Figure 2: 'agglomarated' is not correct in this case; please rephrase. 'Relative abundance of ... children' would refer to the abundance of children not their microbiota composition. Rephrase the whole legend for accuracy. Please add a column of the combined result in City and in Indigenous bcs further down you combine anyway (in the Supplementary figure too); the Venn diagramme takes into consideration all from each location and further down in Figure 3 you combine f&m. Please make C, D, E, F into bars next to each other for direct comparison and if you wish add the numbers (%) too. Further down in Discussion you mention a lot to these Phylum results (309-321) and yet there are no stats on the Phylum results. I think if you are to base so much of your discussion on Phylum you should do stats on these results.

Answer: Thank you for the observation. The new figure legend is ”Figure 2. Relative abundance of microbiota. A) Distribution of bacterial composition (16S rRNA V4) at phylum level of male and female children from Mexico City and the Me’phaa community, phyla with relative abundances < 1 % were included in the “Others”category. B) Venn diagram of ASV 's from Mexico City and the Me’phaa community. C) Relative abundance of GM female children from Me’phaa community D) Relative abundance of GM male children from the Me’phaa community. E) Relative abundance of GM female children from Mexico City. F) Relative abundance of GM male children from Mexico City community.

On the other hand, the goal of this Figure 2 (at Phylum level) was to give a global panorama of the GM composition between populations and sex. We considered that the statistical analyzes should be done at a more specific taxonomic level, as we did in the rest of the manuscript, implementing methods to handle compositional data as deseq2, which detects the fold changes of features, and Zero Inflated regression models. However, your recommendation has been taken into account and for this purpose, we carried out a Chisq analysis to statistically evaluate the proportions found at this taxonomic level between populations and sex. Please see lines 277-292.

Figure 3: The title of this figure and Figure 2 must be rephrased. 'Alpha diversity'is not enough to describe the figure. Also, why A) is so much bigger than B) and C) ? 

Answer:Thank you for your comment. We have changed the title and A) size. Also, we have decided to join Figure 3 and Figure 4

Line 237-239: You state ' The difference...., but not between...' Since you take the whole city (F&M) vs indigenous (F&M) for PD diversity, why don't you show the cumulative diversity in the Figure too rather than keeping m&f separately in A) ? The combined PD should be shown bcs this is what you discuss in the text and then for informative reasons, split into f and m. The whole Figure must be redrawn and for comparisons between the two locations you have to put them in the same box not in separate. Separate boxes would tell the readership that they refer to within-location comparisons. 

Answer: Thank you for the suggestion, we have added the cumulative diversity in figure 2.

Line 237: No need to mention Faith's Index all the time.

Answer:Thank you for your comment. We rephrase this sentence according to this comment. Please see lines:  315-317

Line 252: what doe you mean '... but a tendency' ?

Answer: Thank you. We use tendency for describing the approximation of values  regarding the p.adjusted values. However,  for tendencies, no significant differences are obtained. Therefore, we have changed this sentence according to your assertive comment. Please see lines : 340-341

Figure 4: Please make the title more accurate. Please add PERMANOVA results on the Figure. What we see here is the effect of location?

Thank you for the observation. We have changed the title and added the PERMANOVA results on the Figure. On the other hand, what we are seeing in the figure 4 (now Fig. 3 D and E) is effectively the effect of location, since there was no effect between sexes in both analysis, Please see the results section, lines 315-325 and Figure 3. 

Figure 5: Please make the title more specific. Please specify that here you consider combined M&F from each location. Is that right? 

Thank you for the observation. We have changed the title and added a sentence for clarifying that M&F are combined per location since there was no differences between sexes. Please see lines 361-262 

Figure 6: Specify title. Does the * next to the Family name in each plot show the initial test result or the post-hoc result? 

Thank you for the observation. We have changed the title. The * shows the significant differences (at p< 0.01) between populations. In this case, we did not  perform any post-hoc correction since the only predictor  that was statistically significant was “ population”, which has 2 levels;  “indigenous” and “mexico city”. We reported this in the section of results and clarified in the figure legend. Please see lines:391-392

Lines 309-310: Dear authors, how is it that the fact that the bacteria taxa you mention to be 'common phyla reported for human diversity' is 'interesting' since this is a matter of fact ? Please rephrase bcs the whole sentence message is lost. 

Answer:Thank you. We agree with your observation and we have now deleted “Interestingly”. Please see the new sentence in lines 419-420.

Line 311-312, 312-313: Please specify to which communities you refer to to avoid misconceptions. Please rephrase 'may tell us about...'. Please rephrase the whole 312-313 bcs it does not convey any clear scientific message.

Answer:Thank you. We have rephrased the sentence to “Firmicutes, Bacteroidetes, Actinobacteria and Verrucomicrobia are common phyla reported for human gut diversity in westernized and non-westernized populations.

Also, we replaced “may tell us about” for “which suggests their presence as part of” the general “core” present in the human GM (Fig. 2B). See lines 419-425.

Lines 309-325: Please see comments on Figure 2.

Answer: Yes, we have changed Figure 2.

Lines 367-369: Reference please.

Answer: Thank you. Yes, we have added reference # 21 ( F. A. Ayeni et al., “Infant and Adult Gut Microbiome and Metabolome in Rural Bassa and Urban Settlers from Nigeria,” Cell Rep., 2018, vol. 23, no. 10, pp. 3056–3067, doi: 10.1016/j.celrep.2018.05.018).

 Please see lines 490-492

Lines 377-387: Please rephrase for clarity of take-home message and add a reference in line 379.

Answer: Thank you. We rephrased lines 337-387 and added references in line 379.  Please see lines 502-505.

Lines 394-395: Which are these? Please specify somehow bcs as it is now it reads as if Streptococcus, Clostridium snsu stricto and Erysipelotrichia are not commonly present, which is totally erroneous. Also, how is this intro statement binds/leads to what you discuss further down? Major revision here please (394-401).

Answer: Thank you for the observation. We have rephrased the paragraph. We hope that you find this new version better. Please see lines 521-532.

Lines 399-406, 408: References are needed.

Answer:Thank you for the observation, we have added references in lines 399-408.Please see lines 532-542.

Reviewer 2 Report

Thank you for the point-by-point responses. While I'm eager to see a meta-analysis with these data, it is of course at the authors discretion -- I look forward to reading the follow up works. 

Author Response

We want to thank the reviewer#2 for all the suggestions and contributions to improve our manuscript.We believe that due to the reviewer#2 suggestions, the writing has been improved substantially. We seek to make the comparison between indigenous communities in the near future, so we would love to have his/her point of view for that manusript.